# Digital Transformation of Building Permits: Current Status, Maturity, and Future Prospects

Mariana Ataide, Orjola Braholli and Dietmar Siegele *

Fraunhofer Italia, 39100 Bolzano, Italy; mariana.ataide@fraunhofer.it (M.A.); orjola.braholli@fraunhofer.it (O.B.)
* Correspondence: dietmar.siegele@fraunhofer.it

**Abstract:** Building permits ensure construction meets codes and regulations, but the traditional permitting process is often complex and inefficient. This perspective paper examines the current state and maturity of digitizing and automating building permits. We studied current permit workflows and the recent literature to identify digitization opportunities like online portals, automated code-checking, and data integration. Most jurisdictions are only in the early digital stages, focused on implementing electronic document management and online portals. Some leading cities have piloted more advanced capabilities like automated code compliance checking, but widespread adoption lags. The greatest challenges exist around fragmented IT environments, data integration, organizational inertia, and failing to adapt to technological advancements, such as the example of AI. Achieving higher digital permitting processes requires optimized data sharing, instant feedback loops, and automation-enabled plan reviews. While pockets of innovation exist, mainstream adoption lags behind visionary potential. Realizing the future permitting paradigm demands open data standards, configurable software infrastructure, and organizational commitment to digitize end-to-end. This paper presents regulators and innovators with a perspective framework to evolve permitting towards smarter, faster, and more integrated digital systems and strategies.

**Keywords:** digital building permit; permitting process; maturity model; readiness level; DBP ecosystem; technology





## 1. Introduction

A building permit, in summary, is a permit given by a building authority to construct or renovate a facility. The importance of the process lies in safeguarding building users and regulating the urban fabric. Nevertheless, the dependency of the legal framework and governmental processes makes the process of building permission overly complex, prone to errors, non-transparent, and unpredictably lengthy [1]. Likewise, the architecture, engineering, construction, and operations (AECO) industry is known to be one of the most delayed industries regarding innovation. The digitalization of processes has become one of the most important topics discussed in different domains. Innovative technologies are created in a fast-paced environment where industries need to adopt different techniques to adapt to the new demands on efficiency, accuracy, transparency, and collaboration. The combination of the complex legal framework and the delayed AECO innovations put the digitalization of the building permit process behind other industry' innovations.

Nevertheless, the recent years have shown a positive evolution of technological advancement in the AECO industry. Many studies on building information modeling (BIM), geographic information systems (GISs), digitalization of construction processes, and integration with industry 4.0 have been developed in the past decade [2,3]. The spread of information technology related to building data allowed for big advancements in BIM-enabled model rule-checking. Since Eastman [4] and the definition of the rule-checking steps, many case studies have been presented, and tools have been developed to allow a broader set of rules to be analyzed [5–7]. The rule-based foundation of automated checking

makes the method an essential part of the digitalization of building permits. Therefore, the evolution of automated model checking also enables the evolution of the conversation on digital building permits (DBPs).

Model checking based on data extracted from the model is a largely explored area with many software developments since the broad adoption of BIM in construction. The use of data retrieved from the model to check with the compliance rules allows for consistent information flow throughout the lifecycle of the building asset and reduces the occurrence of errors in the checking process, among other advantages. Since the definition of the four steps for automated rule-checking by [4], there has been a lot of development on the topic. (1) Rule interpretation, (2) model preparation, (3) rule execution, and (4) result reporting each have many studies that explore the matter and allow for the development of each of the phases.

Many software and tools have been developed to cover model preparation, rule execution, and result reporting. Since these are the most straightforward phases, they also have the most potential for automation. Therefore, many studies have focused on the interpretation of regulations and norms. The past few years have provided many studies focused on the implementation of BIM-based DBP. Most of the studies are focused on the technical and technological aspects [6,8–10]. However, as pointed out by [11] in their qualitative study, most of the problems affecting the adoption of BIM in the DBP process rely on organizational factors related to the regulators, while other studies highlight the difficulties of adapting the regulations to machine-readable formats [9,12,13].

Therefore, understanding the maturity level for digitalization in the organizations involved in the DBP process is a key factor for the success of the implementation. A staged roadmap would systematically guide DBP transformation, allowing stakeholders to gauge and incrementally improve focused capabilities. As BIM and GIS models have guided adoption, tailored maturity models can enable progressive advancement towards ambitious DBP goals.

Translating the traditional process into an automated DBP is not a straightforward process. The analysis of regulations, understanding of the workflows within an administration, and creating or translating legal texts into machine-readable format present some of the barriers to full implementation. The current perspective article aims to present a viewpoint on the process to be reached, using BIM and GIS as the data sources and exchange throughout the process lifecycle. By analyzing the literature from the past three years, we provide a possible framework for linking the process of digital building permits with the readiness level to implement it, not only focused on one specific use case but also trying to maximize scalability and address new technological movements that can help accelerate and optimize the process.

Section 2 presents the methodology for this perspective article. Section 3 shows research of the literature on this topic. Section 4 presents a perspective on the DBP process and level of maturity. In Section 5 we show some limitations and future directions. We end with a discussion and conclusion in Section 6.

## 2. Methodology

The research methodology for this paper respects the following structure:

### 2.1. Philosophical Stance and Motivation

This paper takes an analytical approach to understanding experiences and perspectives on the path of digital transformation of the building permit process through the analysis of the recent relevant literature. Afterwards, a qualitative study using document analysis is undertaken to gain insights about the topic.

### 2.2. Data Collection

To collect the latest literature available regarding digital building permits, various scholarly databases, including Scopus, Web of Science, and scientific repositories of univer-

sities were searched for publications. The search involved using combinations of relevant keywords such as "digital building permit", "maturity", "BIM", "GIS", "code compliance", "automated rule-checking", "e-permit", "digital transformation", and "artificial intelligence", "construction digitalization", "3D spatial analysis", "digital tools", "planning permit"," city information modelling", "digital twin", "GeoBIM", and " extended reality". These keywords were chosen to retrieve publications pertaining not only to digital building permits specifically, but also to associated topics, advances, and technologies expected to impact this domain indirectly. The resulting articles were filtered to focus on research published between 2020 and 2023, and they were categorized into subtopics. The relevant articles were selected and analyzed to gain an understanding of the current state and progress made regarding clarity, transparency, and maturity of the digital building permit process, as well as the applied technologies, gap areas, and changing trends. The findings are reported in the following sections.

### 2.3. Literature Research

The literature research entailed a thorough examination of accumulated and filtered publications concerning the building permitting process, the digital transformation taking place in this domain, and associated technologies and events impacting said process. The key takeaways, outstanding challenges, and perspectives on the future were synthesized from this comprehensive review.

In alignment with the focus of this study and based on directly related developments alongside discoveries made during the literature analysis, four primary topic areas were identified to categorize the articles. The subject of building permits and their maturity levels is central to this work, while rule checking and interpretation, emerging technologies, and maturity model types represent complementary topics influencing the core subject. Process, maturity models examining the full process and individual components, technologies, and rule checking constitute the four main topics of this study, providing a framework for additional analysis of the literature moving forward.

To further analyze findings from the literature, six targeted research questions were formulated to address the overarching questions posed in line with the foresight model. Examining the literature through the lens of these precise questions will facilitate analysis of the current situation, contextualize interpretations, and chart future trajectories surrounding the digital transformation of building permits.

### 2.4. Data Analysis

The data analysis was made using qualitative data and manual coding for data analysis. The qualitative method was used to analyze text data by identifying and quantifying key words, themes, or concepts. Classifying data into categories: The data were classified according to the determined topics, to keywords pertaining to each of the topics, to which of the research questions they respond, and to their main focus. Based on that, the literature material was reviewed to understand the current developments and examine the most common applications and implementations adopted across digital building permit transformation stages, specifically where rapid changes are unfolding as departments digitize workflows: the comparative maturity levels across phases and features of digital permit systems; including areas displaying greater versus lagging progression; the existence of strategic frameworks and methodologies guiding incremental advancement across maturity levels for holistic transformation and approaches enabling greater interconnection of capabilities across phases; the key technologies and tools for digital transformation of permits and how they are being utilized within modernizing workflows; the sources of greatest resistance persisting in transitions from manual to digital systems, including contributing organizational, regulatory, and behavioral factors; and future outlooks, and progressive trends envisioned for building permit systems considering technological change and innovation, including how emerging technologies could shape the next phase of transformation.

### 2.5. Perspective and Conclusions

We used the generic foresight model by Voros [14] to analyze our findings, interpret the current situation, and envision prospective futures. Foresight methodologies gather and analyze data to facilitate new ways of thinking about the future, fostering understanding of the past and present as a basis for exploring potential futures [15]. Based on the foresight model by Voros [14], in our research and analysis of the literature, the following questions led the structure of the research: (1) What is going on? What are the literature findings? (2) What seems to be happening? (3) What is really happening? (4) What might happen? (5) What might we need to do? The research method based on the generic foresight model is illustrated in Figure 1.

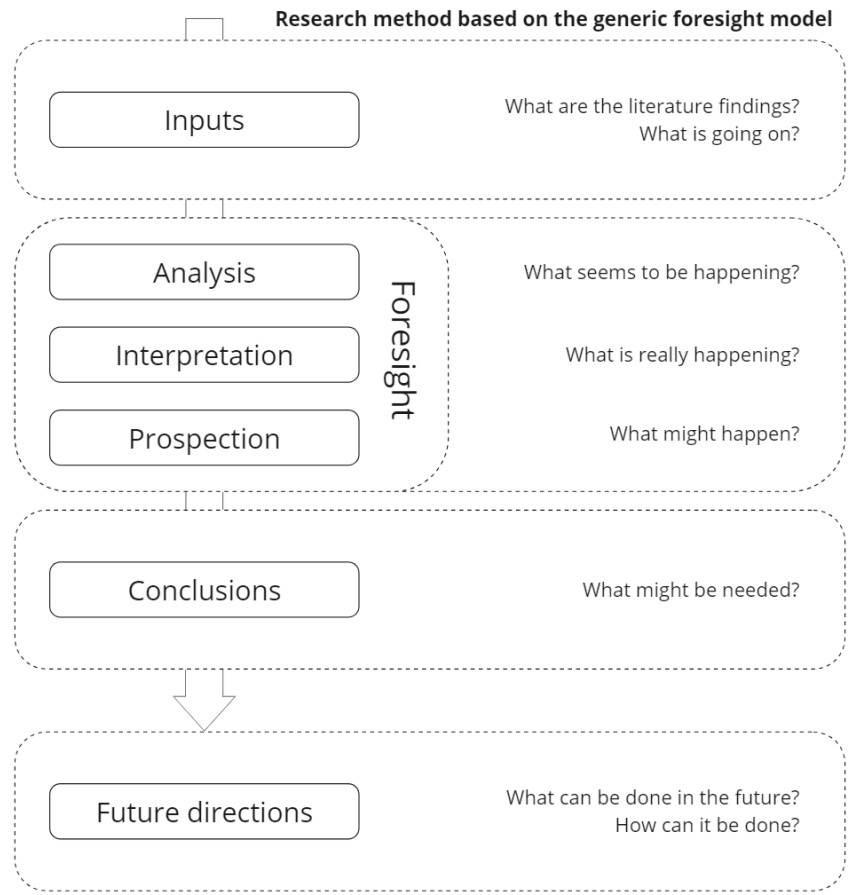

**Figure 1.** Research method based on the generic foresight model by Voros 2003.

This literature review provides a lucid understanding of current developments in the digitalization of permitting processes as well as in other pertinent domains affecting this field. These synthesized findings offer key inputs as a foundation for applying the foresight model to enable further analysis, interpretation, and exploration of future possibilities. The baseline established by reviewing the literature supplies vital context for leveraging the foresight approach to examine the trajectory of digital transformation across different time horizons and scenarios.

A comprehensive technology assessment was conducted to evaluate current information systems underpinning digital permitting, gauging their maturity levels and developmental timelines to project future capabilities. Additionally, a gap analysis compared the present state of digital permitting to an ideal future vision, revealing maturity gaps and areas necessitating innovation.

## 3. Literature Research and Analysis

### 3.1. Lieruature Research

The clear evolution of software tools and of data-driven technologies has paved the way for discussing the digitalization of not only rule-checking, but also of whole process automation, including the digitalization of building permits. Even though the checking of the project is only one part of the permitting process, the automation of this specific part is already a big advancement for increasing transparency and efficiency, being a great advantage for municipalities, applicants, and other stakeholders involved in the building permit.

More recent research has explored the potential for BIM and automation to improve and facilitate the building permit approval process. The work by [16] gives an overview of the literature on DBP, systematizing the studies according to the phases of the process. As pointed out, the most explored phase is related to content review and automated rule-checking. A case study of the Tallinn City Government by [11] revealed important organizational factors that both enable and challenge BIM adoption for building permits, finding that the dynamics differ from general industry BIM adoption. In more specific studies, Ref. [17] developed a web-based prototype that enables automatic 3D modeling and spatial analysis to check proposed building designs against land use regulations and detect conflicts early on, while [18] created optimized checklists to improve the quality of BIM models during detailed design in order to better facilitate approvals, permits, and validation by clients, despite the time limitations often faced. Ref. [19] developed a system called BIMSMACC to semi-automate fire safety code compliance checking in Malaysia using native Autodesk Revit models. Their approach aimed to balance automation with the greater involvement of AEC experts, making it more practical for designers.

Ref. [20] prototyped the automation of common algorithmic tasks in the building permit process using IFC data and computational geometry, demonstrating the general feasibility of digitizing and automating parts of the overall workflow. However, they faced non-technical barriers such as legal issues for practical implementation. Overall, the research indicates that automating portions of the building permit approval process through BIM and algorithms shows promise. However, organizational factors and legal barriers pose challenges to practical implementation. Striking an effective balance between automation and human expert involvement appears important to creating systems that can feasibly be integrated into real-world permit approval workflows. While not yet fully realized, these studies demonstrate progress towards digitizing and streamlining elements of the building permit process through BIM.

Even though the evolution of the conversation on data-driven DBP has gone further, the implementation of real cases and the global view of the process are still lacking. Typically, a building permit involves steps such as pre-consultation, application submission, formal review, content review, permit issuance, and post-construction [21]. Most of the studies are focused on the automation of rule-checking or the translation of specific rules to a machine-readable format, which are directly relatable with the content review phase. However, there is a need to integrate all the phases and have the same data used across the entire process. Recent studies show that a unified solution is far from being achieved, not due to a technical or technological restriction but due to the complexity of the legal frameworks [20].

Digitalization and Readiness Level

The transition from traditional to digital automation is often time-consuming and complex. A recent study by [11] aimed to understand municipalities' challenges in adopting digital building permits. They found organizational factors to be key barriers to implementing a fully digital process. Technology growth often outpaces the organizational and personnel changes needed for its implementation. The necessary transformations for cutting-edge technologies are frequently slow and costly. Thus, a thoughtful plan and a defined system can effectively guide the digital transition.

As organizations pursue digital transformation, maturity models provide critical frameworks to assess and improve readiness. Implementing sophisticated capabilities like digital building permits requires staged progress across dimensions like processes, culture, and staff skills. Maturity models are frameworks for assessing an organization's current processes and capabilities in a domain. They outline an evolutionary progression through maturity levels describing process sophistication, from initial ad hoc states to continuous improvement and optimization. Each level defines the particular process capabilities, best practices, metrics, and competencies characterizing that maturity stage. This encourages organizations to evaluate their capabilities, set improvement goals, and monitor progress. The staged approach accounts for the level of effort and cultural change required to progressively enhance processes, technology integration, and human skills over time.

As BIM and GIS gained prevalence, maturity models were introduced to aid implementation in the AEC sector. Succar [22] proposed a BIM framework defining stages from object-based modeling to integrated project delivery. The BIMMM has since been used to assess BIM maturity. Recent years also brought new perspectives on the topic, with models focusing on aspects from an organizational point of view [22–25] or focusing on the project level [26,27]. Similarly, maturity models for the use of GIS data and the implementation of GIS systems have also been powerful tools for the transition to integrated systems [28,29].

However, the multiplicity of available information often takes a broader perspective than the specifics needed to implement digital building permits. DBPs aim to automate and improve approval through advanced information systems; this requires evolving the processes, systems, integration, capabilities, policies, and culture of the municipality and other partners involved in the process. Aligned maturity models can provide a pathway to digitally transform permitting in a structured way, increasing success chances.

### 3.2. Foresight Analysis and Interpretation

After understanding the state of the art related to efforts and research concerning the digitalization of the permitting process, this section examines the status of the building permit processes, maturity levels, and future outlooks for this digital transformation through a comprehensive review of the latest literature available. Following the principles of the foresight model, we analyzed the most common applications implemented, process maturity, strategic frameworks for advancement, technologies leveraged, change resistance factors, and future trends. Our review reveals that the existing research predominantly focuses on rule-checking functionality and discrete solutions, with fewer studies taking a holistic process view. While the current work provides a baseline understanding of building information modeling integration and compliance tools, gaps persist regarding overarching maturity models, systemic roadmaps, change adoption, and future perspectives. Furthermore, this article synthesizes limitations and opportunities in current knowledge, providing directions for further research on mapping comprehensive digital transformation pathways for modern permit systems. Through this analysis, guided by the following six key questions, we aim to understand maturity phase progression, visualize future scenarios, and delineate a strategic framework for digitizing processes end-to-end. The key questions are as follows:

1. What are the most common applications and implementations being adopted across different stages of the digital building permit (DBP) transformation process, and which of the stages demonstrate the most rapid changes as building departments digitize permit workflows?
2. What is the comparative maturity level across different phases and features of digital building permit systems? Which areas display greater progression versus those lagging in maturity?
3. Do strategic frameworks and methodologies exist to guide building departments in incrementally advancing across different maturity levels for holistic DBP transfor-

mation? What approaches can enable the progression and greater interconnection of digital capabilities across permit process phases?

4. What are the key technologies and tools being leveraged to enable the digital transformation of building permits? How are these technologies being specifically implemented and utilized within modernizing permit workflows?

5. Where does the most resistance persist when transitioning from manual or paper-based building permit processes to digital systems and beyond? What organizational, regulatory, and behavioral factors contribute to inhibiting the adoption of DBP transformations?

6. What are the future outlooks and progressive trends envisioned for building permit systems in an era of exponential technological change and innovation? How could emerging technologies shape the next phase of transformation?

Table 1 shows the distribution of the articles according to the main topics to which they refer and which of the key questions they answer. Based on the scope of this research and the central theme addressed in each article, four salient topics emerged to categorize the literature: process, technologies, rule-checking, and maturity. The publications centered on the overall building permit process and its digital transformations were classified under process. The articles examining maturity models referring to the complete permit process, particular phases, or analogous processes were grouped under maturity. The critical rule-checking phase was a distinct topic given its integral role within permitting workflows. Finally, the literature highlights that technological advancements crucially enable and directly shape the digitization and evolution of permitting systems; hence, technologies formed another major topic. This framework of the four key topics was derived organically from the convergence of the study scope and the predominant focus areas observed across the literature.

**Table 1.** Distribution of studied literature according to topics and key questions.

| Article | Main Topic | Key Question | Reference |
|---|---|---|---|
| A Critical Review of Maturity Model Development in the Digitalisation Era | Maturity, specific phase | 1, 2 | [30] |
| A critical review of text-based research in construction: Data source, analysis method, and implications | Rule-checking | 1, 6 | [31] |
| A Design for Safety (DFS) Semantic Framework Development Based on Natural Language Processing (NLP) for Automated Compliance Checking Using BIM: The Case of China | Rule-checking | 1, 3 | [13] |
| A Multiscale Modelling Approach to Support Knowledge Representation of Building Codes | Rule-checking | 1 | [12] |
| A Perspective on AI-Based Image Analysis and Utilization Technologies in Building Engineering: Recent Developments and New Directions | Technologies | 6 | [32] |
| A Web-based Planning Permit Assessment Prototype: ITWIN4PP | Process | 1, 4, 6 | [17] |
| Adoption of Blockchain Technology through Digital Twins in the Construction Industry 4.0: A PESTELS Approach | Technologies | 4, 6 | [33] |
| An Automatic Process for the Application of Building Permits | Process | 1, 5 | [20] |
| Automated compliance checking in healthcare building design | Process | 1 | [34] |
| Automatic rule-based checking of building designs | Rule-checking | 1 | [4] |
| Automation of Building Permission by Integration of BIM and Geospatial Data | Process | 1 | [35] |
| BIM adoption in the AEC/FM industry—The case for issuing building permits | Process | 1, 3, 4, 5 | [1] |
| BIM for public authorities: Basic research for the standardized implementation of BIM in the building permit process | Process | 1, 5 | [8] |

**Table 1.** *Cont.*

| Article | Main Topic | Key Question | Reference |
|---|---|---|---|
| BIM-Based Automated Code Compliance Checking System in Malaysian Fire Safety Regulations: A User-Friendly Approach | Process | 1, 4, 5 | [19] |
| Building a Next Generation AI Platform for AEC: A Review and Research Challenges | Technologies | 1, 4, 5, 6 | [36] |
| Check and Validation of Building Information Models in Detailed Design Phase: A Check Flow to Pave the Way for BIM Based Renovation and Construction Processes | Process | 1, 4 | [18] |
| Conception, development and implementation of an e-Government maturity model in public agencies | Maturity | 3 | [37] |
| Defining a 'maturity model' in the construction context: A systematic review | Maturity, specific phase | 1, 2, 3, 4 | [38] |
| Development and Implementation of a Maturity Model of Digital Transformation | Maturity, specific phase | 2, 3, 4 | [39] |
| Development of a maturity model for technology intelligence | Maturity, specific phase | 2, 4 | [40] |
| Fire Safety in Tall Timber Building: A BIM-Based Automated Code-Checking Approach Kristina | Rule-checking | 1, 4 | [9] |
| Framework for Automated Model-Based e-Permitting System for Municipal Jurisdictions | Process | 1, 2, 3, 4 | [5] |
| Geobim for digital building permit process: Learning from a case study in Rotterdam | Process | 1, 2, 3, 4, 5 | [2] |
| GIS for the Potential Application of Renewable Energy in Buildings towards Net Zero: A Perspective | Process | 4, 6 | [41] |
| High-level implementable methods for automated building code compliance checking | Process | 1 | [42] |
| Integrated approach for development of automatic building application systems | Rule-checking | 1, 4 | [43] |
| Integrating expertises and ambitions for data-driven digital building permits—The EUNET4DBP | Process | 3, 4, 5 | [44] |
| Optimized decision support for BIM maturity assessment | Maturity, specific phase | 1, 2 | [45] |
| Overview of BIM maturity measurement tools | Maturity, specific phase | 1, 4 | [46] |
| Proposing a methodology to measure and develop BIM maturity in Syria | Maturity, specific phase | 1, 4 | [47] |
| Readiness assessment for BIM-based building permit processes using fuzzy-COPRAS | Maturity | 1, 2, 4 | [48] |
| Research on BIM Application Two-Dimensional Maturity Model | Maturity, specific phase | 1, 2, 4 | [49] |
| The BIM-Based Building Permit Process: Factors Affecting Adoption | Process | 1, 2, 3, 4 | [11] |
| The maturity of maturity model research: A systematic mapping study | Maturity | 1, 2, 4 | [50] |
| Transformer-based approach for automated context-aware IFC-regulation semantic information alignment | Rule-checking | 1, 4 | [51] |
| Translating building legislation into a computer-executable format for evaluating building permit requirements | Rule-checking | 1, 4 | [7] |
| Understanding the Main Phases of Developing a Maturity Assessment Model | Maturity | 1, 3, 4 | [52] |
| Understanding processes on digital building permits—a case study in South Tyrol | Process | 1, 2, 4, 5 | [53] |
| Unveiling the actual progress of Digital Building Permit: Getting awareness through a critical state of the art review | Process | 1, 2, 3, 4, 5 | [16] |
| Integrating disruptive technologies with facilities management: A literature review and future research directions | Technologies | 4, 6 | [3] |
| Augmented Reality for Building Authorities: A Use Case Study in Austria | Technologies | 4, 6 | [54] |

The analysis of the literature reveals that the existing work on the transformation of the DBP process has concentrated predominantly on the rule-checking phase of the process and the application of new technologies. Most papers address key questions 1 and 4, which deal with examining the rule-checking phase and current technologies applied in transforming the permit process digitally (Table 1).

Significant research and practical implementations have focused on data structuring and the integration of BIM to enable automated code compliance checks. In [13], the authors explore how to develop an efficient methodology for the interoperability and semantic representation of data from different sources to enable automated compliance checking of building designs based on BIM. In their study, they propose a natural language processing (NLP)-based semantic framework that implements rules-based automated compliance checking for BIM at the design stage [13]. Furthermore, major construction companies have started developing in-house natural language processing tools to automate text analysis [31]. As NLP technology and digital transformation continue to advance, it is envisioned that automated text analysis will supersede labor-intensive manual tasks in the construction industry [31].

Remaining on rule-checking as a crucial part of the DBP workflow, further findings indicate that more recent studies have centered on this specific phase. In [12], the authors developed a multiscale knowledge model of building codes to enable automated compliance checking, which could help guide designers to include necessary information, reduce gaps between building and regulatory data, and make the compliance process more user-friendly. Refs. [9,51] explore the potential of BIM and IFC for information management in process model compliance checking.

Comparatively, fewer studies respond to key questions 2 and 3. Fewer of the latest studies take a comprehensive view of maturity model development for complete DBP procedures (Table 2), while most of them focus on the maturity of a specific phase of the whole and explore and deploy the maturity topic and maturity models in generic terms. Ref. [48] addresses the maturity of the entire building permit process. The authors assessed readiness for BIM-based building permits in three municipalities using FuzzyCOPRAS and 25 criteria across technology, people, process, and policies [48]. The readiness assessment demonstrated comparative preparedness for digital permitting based on the criteria's status. Refs. [37,50,52] examined the development and implementation of maturity models as a process but did not refer to specific applications in the DBP process.

**Table 2.** Distribution of topics addressed in the studied papers.

| Central Topic | DBP Process | Rule-Checking | Entire Process Maturity | Specific Phase Maturity | Technologies |
|---|---|---|---|---|---|
| % of the studies that address it in the considered literature of the last 3 years | 40% | 20% | 10% | 20% | 10% |

In contrast, other studies on maturity models and readiness refer to specific applications that could potentially integrate into particular phases of the entire DBP process. Ref. [38] focuses on the study of the construction maturity models by establishing a new definition that would facilitate a better understanding among end users in the construction industry. Refs. [39,40] address the maturity topic to fill the gap in the technology adoption field and deepen the understanding of technology intelligence. In addition to the maturity models that refer to specific topics related to the phases of the DBP process, other studies give an overview of the maturity of BIM implementation in various contexts. Ref. [45] fills the gap in BIM maturity assessment methodology by developing an approach to reduce subjectivity and ensure the reliability of evaluation results, an area overlooked in previous studies. Ref. [47] proposes a methodology to measure and develop BIM maturity in general and assess the level of maturity of institutional BIM through the BIMM maturity matrix.

Ref. [49] proposes an innovative BIM maturity model that combines the functions of the project management maturity model (PMMM) and the BIM maturity model.

The aforementioned maturity models could have indirect implications for or be adapted to the DBP process, which is closely tied to decision-making bodies and the construction industry. However, these models do not explicitly address DBP transformation in a direct way. According to [50], the mapping of 237 articles shows that current maturity model research is applicable to more than 20 domains heavily dominated by software development and software engineering.

Although some research covers maturity models for the end-to-end DBP process, there appear to be gaps in devising the strategies or stepwise methodologies required to progress between different maturity levels and interconnect the distinct approval phases more seamlessly.

In addition, most published work involves mapping and analyzing existing manual or semi-digital DBP systems rather than envisaging future outlooks, novel models, and evolving trends in digital permitting. While studies on deploying point solutions are beneficial, there is significant scope for research on holistic process maturity frameworks and strategic roadmaps to digitally transform building permits in a systemic manner. While the majority of studies examined focus on analyzing the DBP process (Table 2), only 20% of them (Table 3) investigate the most resistant areas to change and factors inhibiting adoption, addressing to key question 5.

**Table 3.** Distribution of literature according to whether and how it addresses the research questions in this paper.

| Research Question | 1 | 2 | 3 | 4 | 5 | 6 |
|---|---|---|---|---|---|---|
| % of the studies that address it in the considered literature of the last 3 years | 78% | 33% | 28% | 68% | 20% | 17% |

The literature review highlights a concentration of studies on discrete technologies and compliance checking functionality within DBP systems, compared to work on overarching process maturity, interlinkages, long-term roadmaps, and future ecosystem perspectives, addressing key question 6.

The following discussion in this paper synthesizes key gaps and opportunities to guide further research on the comprehensive digital transformation journey for modern building permit systems.

## 4. DBP Process and Maturity Prospects

### 4.1. The Transitioning Path of the DBP

The mapping of the current process for building permits shows some consistency across phases and steps, even in different countries and urban contexts. Studies such as [19,20] find similarities in the main established phases of building permit processes. Submission, formal review, content review, third-party participation, approval, and issuance are present in almost all analyzed processes, while most municipalities also have pre-submission and post-construction phases (Figure 2). These similarities in the overall process provide a strong starting point for transitioning to a fully automated digital process.

The to-be digital permit process presented in the European Project CHEK Digital Toolkit for DBP [55] represents a transition of the synthesis of comparisons from current as-is processes into a fully digitalized and automated process (Figure 3). The digital process should enable applicants to pre-check their projects prior to submission. This guarantees that the application is consistent with the municipality's requirements and allows the applicant to have an overview of the outcome. Pre-checking increases efficiency by reducing errors in submitted projects while also improving transparency around checking results.

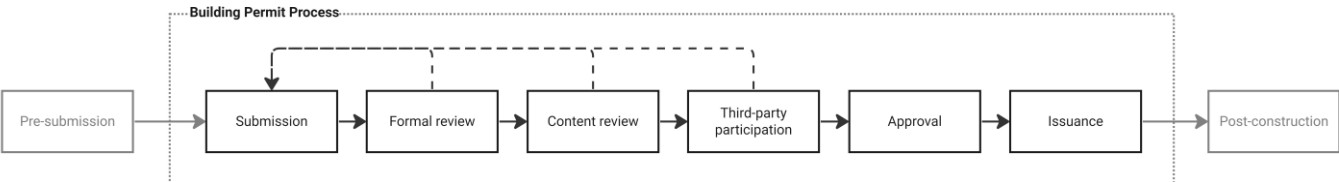

**Figure 2.** Traditional as-is building permit process.

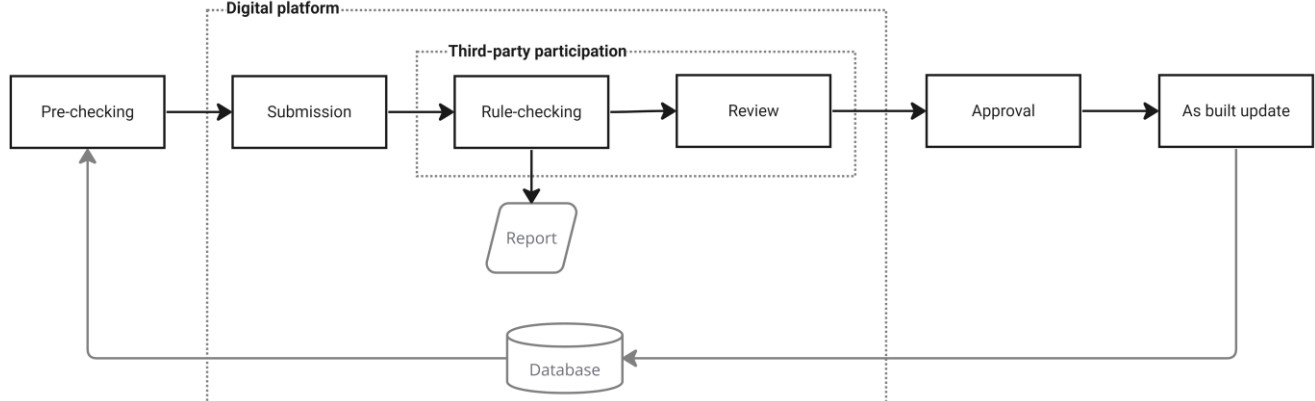

**Figure 3.** To-be DBP process scheme.

The submission is the starting point of the process, where the municipality obtains access to the models and submitted information. A series of checks are performed under the supervision of the municipality's technician, including checking the defined rules for project specifications. As the literature shows, the rule-checking process is nowadays very developed, and there is a large set of rules that can be used to assist municipalities. The rules that have subjective aspects depending on human input should be clearly marked on the rule-checker software, where the technician is able to take a stand on that specific aspect of the analysis. Automated checking is finalized with a report that is available for the consultation of final results.

With the objective of increasing the transparency of the overall process, the steps should be openly visible to the stakeholders involved in the process. Municipalities should be able to grant permission to view and/or comment, while applicants should be able to follow all the steps of their process. The use of BIM and GIS allows for the data exchange during the process to be upright from the start of the process until the stored as-built model. The use of a common environment for all data and stakeholders allows for consistency of data and avoids duplicity.

Many parts of the digitalized process are already achievable through solutions found in research and on the market. However, an integrated ecosystem utilizing consistent data across the full lifecycle is still lacking. This ecosystem could feed post-construction data into the city model. While there are many studies on rule-checking and rule translation, struggles with the implementation of an integrated process persist. These struggles stem from the fact that regulations are often subjective and difficult to translate into machine-readable formats. Various complex interdependencies and references to other norms or clauses are often found when analyzing building projects.

Based on analyzing the process steps and current research, we can identify automated rule-checking as an important part of the overall process. Although the implementation of this step would significantly increase the transparency and efficiency of the DPB process, there are still several other sub-processes that benefit from digitalization advancements, resulting in a fully integrated process and saving time and efficiency for the municipalities, applicants, and stakeholders involved.

The transitioning path of the DBP process presented in this section examines the workflow phases involved in transforming traditional manual building permits to an automated digital process. It maps out the current as-is state with common phases like submission, review, approval, etc., and envisions the ideal to-be digitized process. The transformation journey focuses on the activities and procedures in permitting.

*4.2. Maturity of the DBP Process*

One of the main difficulties in implementing a data-based digital building permit process derives from disparities in capabilities among stakeholders. Since the building authorities issue permits, their organizational readiness to adopt recent technologies is a key factor for successful DBP implementation.

Achieving an optimal level of digital integration requires an elevated maturity level from municipalities, applicants, and other stakeholders involved. To enable automated DBP rollout, a clear implementation plan guiding the transition from current workflows to the desired end state is essential. Maturity models are a valuable tool for assessing capabilities and charting a course based on required competencies. Measuring stakeholders' maturity highlights struggles and areas needing attention.

Adopting digitalized solutions necessitates adjustments to accommodate new processes. Not only for implementing BIM and GIS, but also for emerging technologies that can appear in the future, the ability to efficiently implement new tools depends strongly on the levels of adaptability of organizations and their staff. Influential factors are the organization's current technology, processes, structure, and data quality. Assessing stakeholders' maturity level becomes critical for DBP success; nonetheless, the traditional models available offer little practical utility. Implementation plans should remain flexible to adapt across contexts and innovative solutions. However, lacking a defined scope drastically reduces success chances.

The case studies in the literature on maturity models lean towards single technologies or non-permitting processes [22,46,49,56]. A large gap exists between the adoption of digital permits and the skills municipalities require to implement them. Although the current maturity models offer some utility, most provide limited practical value for holistically improving permitting processes. Additional research should further identify key maturity factors, beyond technology, that facilitate the digital transformation of permitting. Comprehensive models and frameworks addressing these critical knowledge gaps will empower municipalities with the information needed to meaningfully improve.

Based on the cases found in the literature and the analysis of the current permitting processes, four main categories can frame a maturity model for DBP: process, organization, technology, and information (compare Figure 4). Covering the necessary aspects to implement a DBP to-be process, a valuable maturity model should focus on more than just technology since tools change rapidly. The issuer's permitting staff needs robust strategies to adopt innovative solutions capable of leveraging emerging technologies. This requires not just technical knowledge but also adaptability and change management skills.

The comprehensive maturity model framework (Figure 4) accesses those categories by dividing them into capability sets that can be further subdivided into key maturity areas. Each is evaluated from Level 0—Non-existent to Level 5—Optimizing. Each sub-division is rated according to the current state of the organization in that aspect, together with a target-level goal. The gap between the two levels determines the strategy for the organization to adopt in order to achieve the goal.

The main categories have capability sets that measure the maturity of the overall category. The process category is divided into process steps, timeline and transparency of the process, regulatory aspects, and awareness regarding some determined technology. The organizational maturity is measured by the individual and collective knowledge of the staff members, the strategic objectives of the organization, and the capabilities to train internal and external partners. The technology category indicates data management, data analysis, and the interoperability of data and tools. Lastly, the information category is

assessed by the quality of data and information, special capabilities, codes and regulations in machine-readable formats, and the lifecycle of the information workflow.

| CATEGORY | CAPABILITY SET | CAPABILITY LEVELS | | | | | |
|---|---|---|---|---|---|---|---|
| **PROCESS** | Process Steps | Level 0 Non-Existent | Level 1 Initial | Level 2 Defined | Level 3 Managed | Level 4 Integrated | Level 5 Optimizing |
| | Timeline and tranparency | | | | | | |
| | Regulatory | | | | | | |
| | Awareness | | | | | | |
| **ORGANIZATION** | Change readiness | Level 0 Non-Existent | Level 1 Initial | Level 2 Defined | Level 3 Managed | Level 4 Integrated | Level 5 Optimizing |
| | Personal aspect | | | | | | |
| | Strategic objective | | | | | | |
| | Training and external partners | | | | | | |
| **TECHNOLOGY** | Technology for data management | Level 0 Non-Existent | Level 1 Initial | Level 2 Defined | Level 3 Managed | Level 4 Integrated | Level 5 Optimizing |
| | Technology for data analysis | | | | | | |
| | Interoperability and open format | | | | | | |
| **INFORMATION** | Data and standardization | Level 0 Non-Existent | Level 1 Initial | Level 2 Defined | Level 3 Managed | Level 4 Integrated | Level 5 Optimizing |
| | Spacial capability | | | | | | |
| | Codes and regulation | | | | | | |
| | Information workflow | | | | | | |

**Figure 4.** Maturity model framework.

The two first categories (process and organization) are essential for the evolution of technology and information. The interdependencies of capabilities inside the model allow for more cohesive growth. Some capabilities require a certain level of maturity to enable the evolution of others (compare Figure 5). These are milestones that are essential for constructing a staged and efficient roadmap. For example, a city is unlikely to have success in implementing a DBP process if they have good technology for data management but the data feeding the model do not follow a standardized system.

The overall capability assessments guide public and private bodies towards strategic digital permitting improvement. By identifying weaknesses and areas for change across technology, organization, and processes, they can create detailed roadmaps to drive transformation.

Following the transitioning path of the DBP, the maturity model is proposed to evaluate an organization's readiness and capabilities to implement digital building permits. It has four key dimensions—process, organization, technology, and information. Each dimension is broken down into specific capability sets and maturity levels to assess the current state versus desired future state. This framework aims to gauge the preparedness of regulatory bodies to roll out digital permitting solutions.

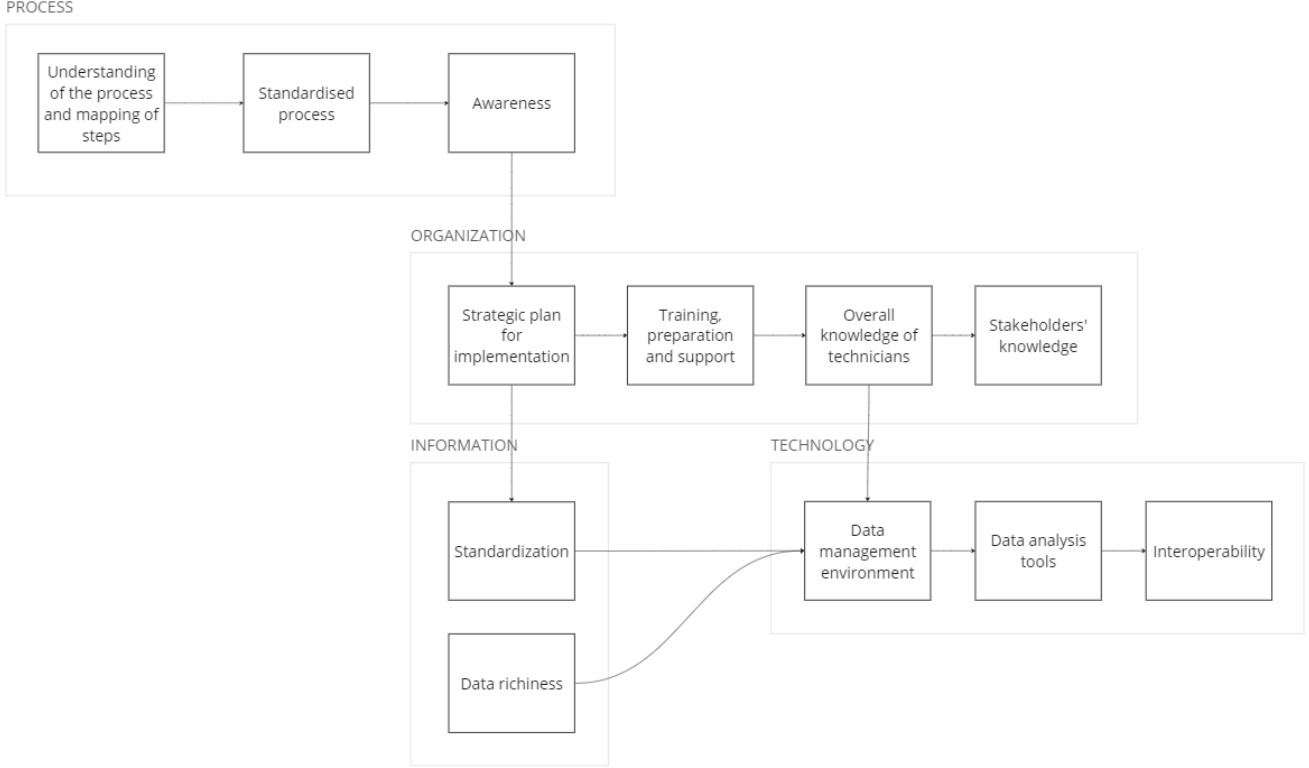

**Figure 5.** Interdependencies of capability sets.

## 5. Future Directions and Limitations

The digitalization of building permit processes has become increasingly relevant for the AECO industry and government administrations, reflecting efforts to optimize and increase transparency. The growth in the literature on digital building permits over the past three years highlights the need to study this topic and adopt technologies already implemented in the private sector. Streamlining building permits through digitalization promises more efficient public processes, benefiting both industry and governance. Despite progress, challenges remain in aligning stakeholders and implementing secure, user-friendly systems. Further research can support the thoughtful adoption of emerging best practices.

The current technologies enable the digitalization of most parts of permitting processes and though solutions remain constrained by available tools, there are already many of them available. Rapid advances in AI raise the threshold for future tech progress across industries, including construction. Studies relating AI and AECO have already been introduced, such as [36], that propose a research framework for the next-generation AI platform integrating these technologies for end-to-end construction management. The perspective paper [32] reviews recent advances in AI-enabled image analysis for building engineering across various applications, identifying progress in photogrammetry, thermography, structural monitoring, and damage detection, along with limitations in software interoperability, fusion, and workflows.

Advanced capabilities in text and image processing offer the potential to automate obstacles like interpreting regulations into machine-readable code [20]. This manual, error-prone step could be expedited by virtual assistants specifically trained for legal translation. Thoughtfully implemented, the automation of regulatory parsing could provide a breakthrough for digital building permits. Rapid advancement in the field of AI may provide a breakthrough for digital building permits. The current challenges in translating rules into machine-readable formats could soon be aided by trained systems for legal frameworks. This could streamline automated rule-checking and other steps of the DBP, unlocking further digital permit process improvements. However, care is needed to audit these systems for hidden biases or misinterpretations. Though incremental, progress in

the responsible adoption of automation may gradually overcome the inherent limitations of manual review at each stage of the permitting workflow. For that, organizations need to increase their overall maturity and of their personnel so they can rapidly adapt to the new processes.

However, based on the findings of the analysis conducted in this study, it is evident that the research focusing on the future prospects of the digital building permit evolution, specifically in light of technological advancements, is relatively limited compared to other topics of investigation. The importance of aligning technological capabilities with advancements cannot be overstated. To fully leverage the potential benefits and possibilities offered by digital building permits, it is crucial to ensure that technological progress is accompanied by the necessary qualifications. This entails developing and implementing innovative solutions that can effectively integrate and harness the evolving technologies. By recognizing the significance of this symbiotic relationship between technological advancements and capabilities, stakeholders can pave the way for a more seamless and successful evolution of the digital building permit system. Further research and exploration in this domain are warranted to fully comprehend the implications and opportunities that arise from this convergence.

## 6. Discussion and Conclusions

This study focused on reviewing the digital building permit and the level of maturity of the process. The literature research analyzed recent findings on digital building permits and process maturity frameworks to understand the current state and future outlook for digitally transforming permit workflows. The analysis reveals that most studies have concentrated on automated rule-checking, applying technologies like BIM to enable compliance. Comparatively less work is required to develop a comprehensive process view or maturity models to incrementally guide stakeholders towards full digital integration. While automating discrete phases is beneficial, further research is needed on interconnected frameworks, ecosystems, and strategic roadmaps to digitally transform the permitting process.

Our key findings show that the rule-checking phase has undergone significant modernization through data structuring and integration to power automated compliance. However, progress is uneven, as organizational readiness lags behind technical capabilities. Comprehensive process maturity models and guidelines tailored to building authorities are lacking. Nevertheless, emerging technologies could provide breakthroughs. AI-enabled text analysis may help automate complex regulatory reviews. To fully realize digital transformation, stakeholders need high maturity not only in technology and data but also in organization and processes. The ability to strategically implement innovative technologies and adapt to rapidly changing trends is a critical enabler for organizations to successfully embrace the digital era.

Delving further into the research, we formulated six key questions to conduct a qualitative analysis of the most recent literature contributions. The primary objective of these questions is to provide a comprehensive understanding of the ongoing advancements in the field of digital building permits and assess the level of maturity achieved this far. Additionally, we seek to envision future directions and potential pathways for further development and implementation of digital building permit processes. By addressing these questions, we aim to contribute to the broader discourse on the subject and shed light on the current state and prospective advancements in this domain. Accordingly, the result of the analysis based on the six key questions can be summarized as follows:

1.　The literature review reveals that the most common applications and implementations adopted for DBP transformation are centered around rule-checking and BIM integration for automated code compliance. The majority of studies focus on these areas, which indicates that the rule-checking phase is undergoing rapid change and innovation in the digitalization of workflows. However, fewer studies examine the end-to-end digitization process.

2. Significant gaps appear in research and frameworks assessing the comparative maturity levels across different phases of the digital permit system holistically. Most maturity model research concentrates on specific aspects rather than taking a comprehensive view of the whole process. Although some work examines BIM and technology adoption maturity, maturity assessments for integrated DBP systems are limited.

3. This review finds limited strategic frameworks and methodologies that could guide private or public bodies in incrementally advancing through maturity levels for complete DBP transformation. There is a lack of stepwise approaches or interconnectivity between distinct phases of DBP, pointing to an opportunity for mapping systematic progression.

4. The key technologies leveraged are in the implementation and integration of BIM, GIS, and IFC for automated rule-checking and information management. Construction companies are also developing NLP tools for text analysis automation. However, studies that take a wider view of emerging technologies are scarce.

5. Only a small percentage of later studies empirically examine organizational resistance, indicating a gap. Research is needed to identify where the most persistent resistance lies when transitioning to digital systems and the behavioral factors that drive inhibition to change adoption.

6. The literature predominantly focuses on analyzing current tools rather than future outlooks. There is a significant opportunity for studies envisioning progressive trends, next-generation systems, and long-term roadmaps or maturity trajectories for DBP ecosystems enabled by new technologies.

This study mapped the current digital permit landscape, revealing gaps and opportunities for further system integration. While technical solutions are maturing, strategies and frameworks to unite components into a cohesive workflow are needed. This study also presents an overview of a framework for a maturity model dedicated to the implementation of a DBP. Maturity models and roadmaps tailored for building authorities will provide invaluable tools to guide this modernization. Digital transformation holds immense potential to rapidly improve permit efficiency, quality, and transparency. Future research can enable the construction of robust digital ecosystems where consistent information seamlessly flows across all participants during the complete process lifecycle, from the pre-application until the as-built data.

In conclusion, the main contributions of this paper include providing a comprehensive literature review and analysis on the digital transformation of building permits, synthesizing key developments, gaps, and future outlooks in this domain. This paper examines the current status, maturity levels, and progression of digital capabilities across different phases of building permit systems based on extensive mapping of the literature. It proposes a framework for a maturity model tailored to assess and guide the implementation of digital building permits across process, organization, technology, and information dimensions. This paper analyzes research progress through six key questions to highlight opportunities in strategic frameworks, interconnectivity, change management, future technologies, and ecosystem perspectives. It discusses the limitations of the current literature concentrated on rule-checking and discrete solutions rather than holistic process maturity and roadmaps. This paper recommends future research directions such as organizational readiness, emerging technologies like AI, and envisioning next-generation permit systems and long-term trajectories. In summary, its core values lie in its comprehensive analysis based in the literature, proposed maturity model framework, research-questions-driven assessment, and delineation of gaps and future outlooks to guide advancement in digitally transforming building permits.

**Author Contributions:** Review of the literature, conceptualization of the maturity model, and writing, M.A. and O.B.; writing revision and work supervision, D.S. All authors have read and agreed to the published version of the manuscript.

**Funding:** Funded by the European Union, grant number 101058559—CHEK. The views and opinions expressed are, however, those of the author(s) only and do not necessarily reflect those of the European Union. Neither the European Union nor the granting authority can be held responsible for them.

**Data Availability Statement:** Not applicable.

**Conflicts of Interest:** The authors declare no conflict of interest.

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
