# Peer review of "Digital Transformation of Building Permits: Current Status, Maturity, and Future Prospects"

_buildings, doi:10.3390/buildings13102554_

Round 1
Reviewer 1 Report
This research paper addresses the future direction of building permits.
However, an overall revision is imperative to articulate the paper's core content effectively.
A structural revision emphasizing the thesis's key findings is necessary, as the core values intended to be conveyed are not clear.
[Section 1]
- Lines 49~51: The authors' claims in this sentence rely solely on reference 1. To fortify the argument's logic, it is essential to cite various references.
- Lines 52~54: It is crucial to include citations from numerous past studies, particularly those focusing on technical aspects.
[Sections 2 & 3]
- Section 3 appears to contain citations supportive of Section 1, suggesting it could serve as a literature review.
- Currently, Section 2 pertains to methodology, but it lacks prior research on data collection or analysis. Consider rearranging Sections 2 and 3.
[Section 4]
- The citations underpinning the DBP framework in Sections 1, 2, and 3 seem inadequate.
- Clarify the basis for deriving the main topic and key question in Table 1.
- Review the title of Section 4 as it appears to describe a process for constructing a DBP framework through confirmation of existing research.
[Section 5]
- Elaborate on the distinction between the DBP framework in Section 4 and the DBP process in Section 5.
- Explain the relationship between the maturity model framework and the DBP framework depicted in Figure 1.
Reviewer 2 Report
Intelligent construction is the main development direction of construction engineering in the future. This paper explores the current status and maturity of digitization and automation of building permits, which can provide support for intelligent construction. Suggest publishing the paper after modification. The opinions are as follows:
1.The chapter structure of this article needs to be reorganized, for example, the content in the second part is too few.
2.There are few images in the paper. It is recommended to add more images to increase readability.
3.The format of the paper needs to be unified, such as line 267.
4.How is the method proposed in this article implemented? What is the effect of the application scenario?
Minor editing of English language required.
Round 2
Reviewer 1 Report
The authors addressed all my comments well.